# Pathogenic *TNNT1* variants are associated with aberrant thin filament compliance and myofibre hyper-contractility

Jenni Laitila[1,2] (iD), Christopher T. A. Lewis[1], Anthony L. Hessel[3,4] (iD), Guido Primiano[5,6] (iD), Aurelio Hernandez-Lain[7,8], Chiara Fiorillo[9], Michael W. Lawlor[10], Coen A. C. Ottenheijm[11], Heinz Jungbluth[12,13], Ka Fu Man[14] (iD), Arianna Fornili[14] and Julien Ochala[1] (iD)

[1]*Department of Biomedical Sciences, University of Copenhagen, Copenhagen, Denmark*
[2]*Folkhälsan Research Center, Helsinki, Finland and Department of Medical Genetics, University of Helsinki, Helsinki, Finland*
[3]*Institute of Physiology II, University of Muenster, Muenster, Germany*
[4]*Accelerated Muscle Biotechnologies, Boston, Massachusetts, USA*
[5]*Fondazione Policlinico Universitario Agostino Gemelli IRCCS, Rome, Italy*
[6]*Dipartimento di Neuroscienze, Università Cattolica del Sacro Cuore, Rome, Italy*
[7]*Neuropathology Unit, Department of Pathology, 12 de Octubre University Hospital, Madrid, Spain*
[8]*4imas12 Research Institute, Rare Diseases Network Biomedical Research Center (CIBERER), 12 de Octubre University Hospital, Madrid, Spain*
[9]*Child Neuropsychiatry, IRCCS Istituto G. Gaslini, DINOGMI University of Genova, Genova, Italy*
[10]*Diverge Translational Science Laboratory and Medical College of Wisconsin, Milwaukee, Wisconsin, USA*
[11]*Amsterdam UMC Location Vrije Universiteit Amsterdam, Physiology, De Boelelaan, Amsterdam, Netherlands*
[12]*Randall Centre for Cell and Molecular Biophysics, Muscle Signalling Section, Faculty of Life Sciences and Medicine, King's College London, London, UK*
[13]*Department of Paediatric Neurology, Neuromuscular Service, Evelina London Children's Hospital, Guy's and St Thomas' Hospital NHS Foundation Trust, London, UK*
[14]*Department of Chemistry, School of Physical and Chemical Sciences, Queen Mary University of London, London, UK*

Handling Editors: Peying Fong & Kevin Murach

The peer review history is available in the Supporting Information section of this article (https://doi.org/10.1113/JP288109#support-information-section).

After an MSc in Human Genetics, **Jenni Laitila** started her work on one of the most common congenital myopathies, i.e. nemaline myopathy, together with Professor Carina Wallgren-Pettersson and Katarina Pelin (Folkhälsan Research Center, Helsinki) and later with Professor Nigel Laing (University of Western Australia). She obtained her PhD in 2019 and continued working on nemaline myopathy (due to *TNNT1* mutations) and potential therapeutic interventions with Dr Julien Ochala (University of Copenhagen). Jenni has now established a new team as part of the Myofin group at the Folkhälsan Research Center, focusing on the molecular and cellular changes leading to congenital myopathies, uncovering novel molecular targets for therapies.

The Journal of Physiology

**Abstract figure legend** Graphical representation of the main results. The cartoon indicates that, in the presence of *TNNT1* variants, the thin filament is more compliant, and more easily activated leading to higher myofibre $Ca^{2+}$ sensitivity and cellular hyper-contractility.

**Abstract** In skeletal muscle, troponin T (TnT) exists in two isoforms, slow skeletal TnT (ssTnT) and fast skeletal TnT (fsTnT), encoded by the *TNNT1* and *TNNT3* genes, respectively. Nonsense or missense *TNNT1* variants have been associated with skeletal muscle weakness and contractures and a histopathological appearance of nemaline myopathy (NM) on muscle biopsy. Little is known about how *TNNT1* mutations ultimately lead to muscle dysfunction, preventing the development of targeted therapeutic interventions. Here, we aimed to identify the underlying molecular biophysical mechanisms, by investigating isolated skeletal myofibres from patients with *TNNT1*-related NM as well as from controls through a combination of structural and functional assays. Our studies revealed variable and unusual ssTnT and fsTnT expression patterns and post-translational modifications. We also observed that, in the presence of *TNNT1* variants, the thin filament was more compliant, and this was associated with a higher myofibre $Ca^{2+}$ sensitivity. Altogether, our findings suggest TnT remodelling as the key mechanism ultimately leading to molecular and cellular hyper-contractility, and then inhibitors of altered contractility as potential therapeutic modalities for *TNNT1*-associated NM.

(Received 13 November 2024; accepted after revision 15 April 2025; first published online 23 April 2025)

**Corresponding author** Julien Ochala: Department of Biomedical Sciences, University of Copenhagen, Copenhagen, Denmark. Email: julien.ochala@sund.ku.dk

**Key points**

- No therapeutic treatment exists for patients with genetic *TNNT1* mutations and skeletal muscle weakness/contractures.
- In these patients, expression and post-translational modifications of troponin T are severely disrupted.
- These are associated with changes in thin filament compliance where troponin T is located.
- All these induce muscle fibre hyper-contractility that can be reversed by mavacamten, a myosin ATPase inhibitor.

## Introduction

In skeletal muscle, troponin T (TnT) exists in two isoforms, slow skeletal muscle TnT (ssTnT) and fast skeletal muscle TnT (fsTnT), encoded by the *TNNT1* and *TNNT3* genes, respectively. In 2000, a pathogenic *TNNT1* founder variant characterized by a stop codon in exon 11 was identified in the North American Amish community, associated with a form of congenital myopathy named nemaline myopathy (NM) type 5 (NEM5, MIM 605355) on muscle biopsy (Johnston et al., 2000). Since then, additional recessively inherited nonsense or missense *TNNT1* mutations have been related to NM in non-Amish patients of different ethnicities (Abdulhaq et al., 2016; Fattahi et al., 2017; Konersman et al., 2017; Marra et al., 2015; van der Pol et al., 2014). *TNNT1*-related NM patients reported to date often display early-onset symptoms characterized by hypotonia and tremors, weakness of the limb and respiratory

muscles as well as multiple contractures (Fox et al., 2018). Although *TNNT1*-related NM has been recognized for some time, little is known about the underlying biochemical mechanisms by which the causative *TNNT1* variants ultimately lead to muscle dysfunction. Hence, no cure is currently available, and management is largely symptomatic.

Based on studies in transgenic mouse models, some of the *TNNT1* pathogenic variants, such as the original Amish founder mutation featuring a stop codon in exon 11, induce a partial or complete loss of ssTnT with truncated forms being unable to incorporate properly into the myofilaments and being degraded rapidly (Wang et al., 2005). Such loss of ssTnT is thought to cause preferential atrophy and degeneration of slow twitch myofibres (Abdulhaq et al., 2016; Fattahi et al., 2017; Johnston et al., 2000; Konersman et al., 2017; Marra et al., 2015; van der Pol et al., 2014; Wei et al., 2014). These detrimental cellular mechanisms are definite contributors to the

muscle weakness and contractures seen in the patients; however, the pathology is likely to be more complex.

In skeletal muscle, the regulation of muscle contraction involves structural changes in both thick and thin filaments. Thin filaments are copolymers composed of actin subunits, tropomyosin and the troponin complex. According to the steric model of myofilament activation, in the absence of $Ca^{2+}$, tropomyosin sterically hinders interactions between actin and myosin molecules. Upon addition of $Ca^{2+}$, its binding to troponin C results in TnT conformational changes and azimuthal movement of tropomyosin around the actin filament to unmask binding sites on actin for the myosin motor, allowing cross-bridge formation, force production and motion (Gordon et al., 2000). TnT is then crucial for thin filament activation and normal myofilament function. Hence, in the present study, besides inducing a partial or complete loss of ssTnT, we initially hypothesized that *TNNT1* pathogenic variants would disrupt the regulation of the troponin complex, resulting in accrued muscle fibre force production and ultimately contributing to muscle symptoms. To test this hypothesis, we isolated skeletal myofibres from patients with *TNNT1*-related NM as well as normal controls, and performed a combination of proteomics, molecular dynamics simulations, X-ray diffraction and cell mechanical studies.

## Materials and methods

### Ethics approval

All human tissue, as part of the normal diagnostic process, was consented, stored and used in accordance with the Human Tissue Act, UK, under local ethical approval (REC 13/NE/0373) and conformed to the standards set by the *Declaration of Helsinki*.

### Human subjects

Quadriceps muscle biopsy specimens were obtained from eight patients diagnosed with NM together with pathogenic variants in the *TNNT1* gene, and from six controls with no history of neuromuscular disease. Some of these samples were obtained from the Congenital Muscle Disease Tissue Repository at the Medical College of Wisconsin. Details of all the patients and controls are presented in Table 1. All the muscle biopsy specimens were snap-frozen and stored at −80°C until analysed.

### Identification of phosphorylation/acetylation on ssTnT and fsTnT

For a detailed description of protein separation (SDS-PAGE gel), band excision, tryptic in-gel digest, peptide separation and sequencing, please see Sonne et al. (2023). Raw MS data were analysed with MaxQuant (v1.6.15.0) and phosphorylation (STY) or acetylation (K) sites analysed. The MS proteomics data have been deposited to the ProteomeXchange Consortium via the PRIDE partner repository with the dataset identifier PXD054504 (Perez-Riverol et al., 2022).

### Molecular modelling and simulation

A structural model of the human skeletal thin filament junction with the fsTnT segments containing the Lys58 residue was built through homology modelling using the cryo-electron microscopy structure of the porcine cardiac junction in the relaxed state (PDB ID: 8DD0; Risi et al., 2023) as template. Our model (see Fig. 2A) contains two segments of fsTnT (UniProt ID: P45378-7, residues 36–112), six subunits of human alpha skeletal actin (UniProt ID: P68133, residues 7–377) and eight segments of human alpha-1 tropomyosin (UniProt ID: P09493-1, residues 1–70 for the N-terminal segments and 230–284 for the C-terminal ones). Target–template sequence alignments were performed with Praline (Simossis & Heringa, 2005). Sequence identity values (calculated over the residue intervals indicated above) were 74% for fsTnT, 99% for actin, and 96% (N-terminal segments) and 100% (C-terminal segments) for tropomyosin. The ADP molecules and $Mg^{2+}$ ions bound to the actin sub-units in the template were retained. A total of 300 models was generated with MODELLER (Sali & Blundell, 1993) (version 10.3). The model with the best DOPE score was selected as the starting structure for the simulations indicated as wild-type (WT) in the text. The quality of the model after energy minimization (see below) was evaluated through MolProbity (Williams et al., 2018) (MolProbity score = 2.06) and QMeanDisCo (Studer et al., 2020) (global score = 0.79). The Lys58Gln mutation designed to mimic the effect of Lys acetylation was introduced in both fsTnT segments using ModLoop (Fiser & Sali, 2003) and the resulting model was used as the starting point for the mutant simulations.

Molecular dynamics (MD) simulations were performed with GROMACS (version 2022.4) and the Amber99SB*-ILDN force-field. The ADP molecules bound to actin were described using parameters from the literature (Meagher et al., 2003). The protonation state of ionizable residues at pH 7.0 was determined with PROPKA (version 3.4.0). Systems were solvated with a truncated octahedron box of TIP3P water molecules. A minimal distance of 12 Å was set between the protein and the walls of the box. Counterions ($Na^+$ and $Cl^−$) were added to neutralize the systems and to set the overall ionic strength to 50 mM, leading to a total of ∼1.04 million atoms. Periodic boundary conditions were applied. The

**Table 1. Patient and control muscle biopsy samples used**

| Age | Sex | Genotype | Disease | Sub-classification |
|---|---|---|---|---|
| 49 years | Female | Homozygous *TNNT1* (c.786delG exon 13) p.(Lys263Serfs*36) | Late-onset mild nemaline myopathy (Petrucci et al., 2021) | Mild |
| 51 years | Male | Homozygous *TNNT1* (c.551_552delinsCA) p.(R184P) | Late-onset mild nemaline myopathy (Martin-Jimenez et al., 2022) | Mild |
| 37 years | Male | Homozygous *TNNT1* (c.311A>T exon 9) p.(E104V) | Mild nemaline myopathy (Konersman et al., 2017) | Mild |
| 12 years | Male | Homozygous *TNNT1* (c.311A>T exon 9) p.(E104V) | Mild nemaline myopathy (Konersman et al., 2017) | Mild |
| 10 months | Female | Homozygous *TNNT1* (c.661G>T exon 12) Nonsense variant p.(E221X) | Severe nemaline myopathy (D'Amico et al., 2019) | Severe |
| 1 year | Female | Homozygous *TNNT1* (c.505C>T exon 11) Nonsense variant p.(E180X) | Severe nemaline myopathy (Fox et al., 2018) | Severe |
| 1 year | Female | Homozygous *TNNT1* (c.505C>T exon 11) Nonsense variant p.(E180X) | Severe nemaline myopathy (Fox et al., 2018) | Severe |
| 1 year | Male | Homozygous *TNNT1* (c.574_577delinsTAGTGCTGT) Nonsense variant p.(L203X) | Severe nemaline myopathy (Abdulhaq et al., 2016) | Severe |
| 55 | Male | – | – | – |
| 37 | Male | – | – | – |
| 49 | Male | – | – | – |
| 44 | Female | – | – | – |
| 21 | Female | – | – | – |
| 35 | Female | – | – | – |

The patients have been separated into two sub-categories according to their phenotype (column to the right). Note that severe cases were significantly younger than mild cases and controls.

Particle Mesh Ewald method was used for electrostatic interactions, with a 9 Å cutoff for direct space sums, a 1.2 Å fast Fourier transform grid spacing and a fourth order interpolation polynomial for the reciprocal space sums. A 9 Å cutoff was used for van der Waals interactions and long-range corrections to the dispersion energy were included. Constraints were applied to all covalent bonds with hydrogen atoms in the protein using LINCS.

Energy minimization, equilibration and production simulations were run following a similar protocol as that previously used for simulations of myosin (Antonovic et al., 2023; Hashem et al., 2017), with the following variations. A 1 fs time step was used for the initial stages of the equilibration and increased to 2 fs for the last equilibration stages and the production. The total length of all the equilibration stages was 49 ns. Positional restraints were applied to limit the motion of backbone atoms at the truncated termini of tropomyosin (residues 55–70 and 230–245) and troponin (residues 36–38 and 93–112). Residue intervals were determined in preliminary calculations as having root mean square fluctuation values averaged over three unrestrained simulations >5 Å.

Five replicas were run for both the WT and the mutant system, with production runs of 500 ns, for a total of ∼5 μs. Snapshots were saved every 100 ps. The root mean square deviation (RMSD) of all $C_\alpha$ atoms from the minimised structure was computed with MDAnalysis (version 2.4.2) to monitor convergence of the simulations. The hydrogen bond (HB) analysis was performed with the VMD Hbonds plugin (version 1.3), using a threshold of 3.5 Å on the HB distance and of 30° on the HB angle. The contact map was computed with the Contact Map Explorer python package (version 0.7.0). Two residues were considered to be in contact if the minimum distance calculated over all pairs of non-hydrogen atoms was less than 4.5 Å. The *gromos* clustering algorithm implemented in GROMACS was used to cluster frames from the concatenated production

trajectories (snapshots saved every 1 ns). A cut-off of 3 Å was used on the distance between frames, measured as the RMSD calculated over residues forming selected HB pairs (heavy atoms only). Cluster centroids or the closest frames in time from the same cluster showing the HB of interest were chosen as representative structures.

## Solutions

As previously published (Ross et al., 2019), the relaxing and activating solutions contained 4 mM Mg-ATP, 1 mM free $Mg^{2+}$, 10 mM imidazole, 7 mM EGTA, 14.5 mM creatine phosphate, and KCl to adjust the ionic strength to 180 mM and pH to 7.0. Additionally, the concentrations of

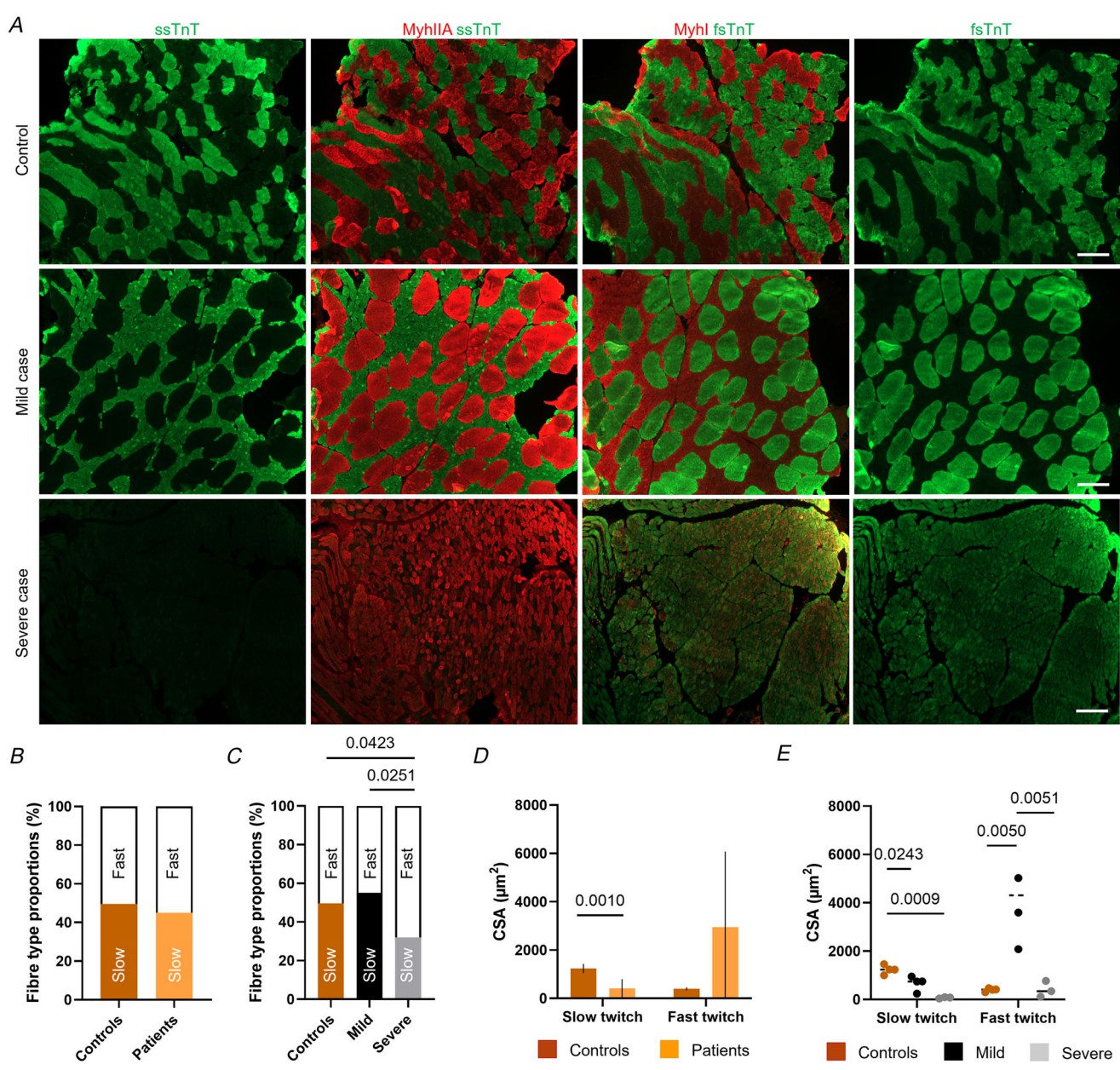

**Figure 1. Histological abnormalities**
*A*, representative muscle cross-sections from one control and two patients. These patients classified as mild or severe (Table 1) have varying expressions of ssTnT and fsTnT in slow twitch (MyhI) and fast twitch (MyhIIA) muscle fibres. Magnification, 10×; scale bar = 100 μm. *B* and *C*, fibre type proportions are displayed for controls and for all pooled patients (*B*) or for patients separated according to their phenotypes: mild or severe (*C*). *D* and *E*, similarly, cross-sectional areas (CSAs) for both slow and fast twitch fibres are presented for controls and pooled patients (*D*) or mild/severe cases (*E*). Mean, standard deviations and/or individual data points (circles) are shown. Two-way ANOVAs with Tukey's *post hoc* test were applied to compare groups (level of significance $P < 0.05$). [Colour figure can be viewed at wileyonlinelibrary.com]

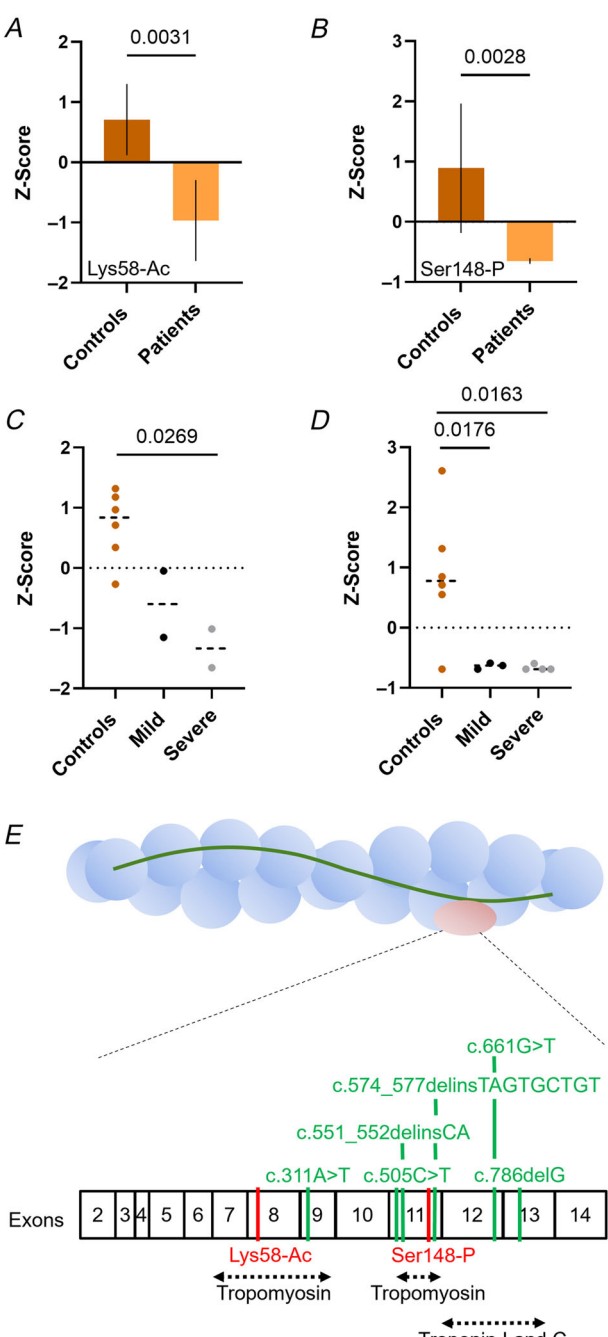

**Figure 2. TnT post-translational modifications**
*A–D*, the Z-score for two specific post-translational modifications, Lys58-Ac and Ser148-P. Mean, standard deviations and/or individual data points (circles) are presented. Unpaired *t* tests with Welch correction (*A* and *B*) or one-way ANOVAs with Tukey's *post hoc* test (*C* and *D*) were used to compare groups (level of significance $P < 0.05$). *E*, an overview of the mutations (in green) and post-translational modifications (in red) along the sequence/exons of TnT. Regions where tropomyosin as well as troponin I and C bind are also shown. [Colour figure can be viewed at wileyonlinelibrary.com]

free $Ca^{2+}$ were $10^{-9.0}$ M (relaxing solution, *p*Ca 9.0) up to $10^{-4.5}$ M (maximum activating solution, *p*Ca 4.5).

## Muscle preparation and fibre permeabilization

Cryopreserved human muscle samples were immersed in a membrane-permeabilizing solution (relaxing solution containing glycerol; 50:50 v/v) for 24 h at −20°C, after which they were transferred to 4°C and bundles of ∼50–100 muscle fibres were dissected free. These bundles were kept in the membrane-permeabilizing solution at 4°C for an additional 24 h (to allow for a proper skinning/membrane permeabilization process). After these steps, bundles were stored in the same buffer at −20°C for use up to 1 week (Ross et al., 2019, 2020).

## X-ray diffraction recordings and analyses

On the day of the experiments, bundles were placed in a plastic dish containing the relaxing solution. They were then transferred to the specimen chamber, filled with the relaxing buffer. The ends of these thin muscle bundles were then clamped at a resting sarcomere length (≈2.20 μm). Subsequently, X-ray diffraction patterns were recorded at 15°C using a CMOS camera (Model C11440-22CU, Hamamatsu Photonics, Sizuoka, Japan, 2048 × 2048 pixels) in combination with a 4 inch image intensifier (Model V7739PMOD, Hamamatsu Photonics). The exposure time was 500 ms. The X-ray wavelength was 0.10 nm and the specimen-to-detector distance was 2.14 m. For each preparation, ∼20–30 diffraction patterns were recorded at the BL40XU beamline of SPring-8 and were analysed as described previously (Ochala et al., 2010). To minimize radiation damage, the exposure time was kept short and the specimen chamber was moved by 100 μm after each exposure. Following X-ray recordings, background scattering was subtracted, and the equatorial as well as the major myosin meridional reflections were determined as described elsewhere previously (Li et al., 2015; Ochala et al., 2010).

## Single muscle fibre force production

Muscle fibres were dissected in the relaxing solution. They were then individually mounted between connectors leading to a force transducer (model 400A; Aurora Scientific, Aurora, ON, Canada) and a lever arm system (model 308B; Aurora Scientific). Sarcomere length was set to ≈2.50 μm and the temperature to 15°C (Ochala et al., 2021). Fibre cross-sectional area (CSA) was estimated from the width and depth, assuming an elliptical circumference. To determine the maximal isometric force, myofibres were immersed in the activating solution. The specific force reported in the Results section

corresponds to absolute force normalized to myofibre CSA. To avoid misinterpretation due to the type of myosin heavy chain, for the mechanical measurements, we assessed the sub-types using immunofluorescence staining as previously described (Ochala et al., 2021). Briefly, flow-chamber mounted myofibres were stained with an anti-$\beta$-cardiac/skeletal slow myosin heavy antibody (MYH7, IgG1, A4.951, sc-53090 from Santa Cruz Biotechnology, Santa Cruz, CA, USA, dilution: 1:50). Myofibres were then washed in PBS/0.025% Tween-20 and incubated with a secondary antibody: goat anti-mouse IgG1 Alexa 555 (from ThermoScientific, Waltham, MA, USA, dilution 1:1000) in a blocking buffer. After washing, muscle fibres were mounted in Fluoromount. To identify the type of fibres (slow twitch *versus* fast twitch), images were acquired using a confocal microscope (Axiovert 200, 63× oil objective, Zeiss, Oberkochen, Germany) equipped with a CARV II confocal imager (BD Biosciences,

Franklin Lakes, NJ, USA) (Ross et al., 2019, 2020). Pure or hybrid fibres positive for MYH7 were considered slow twitch whilst negative myofibres were defined as fast twitch (Laitila et al., 2024).

## Histology

Immunolabelling was performed on 10 μm cryosections, fixed in 4% paraformaldehyde (10 min), permeabilized in 0.1% Triton X-100 (20 min) and blocked in 10% normal goat serum (50062Z, Life Technologies, Carlsbad, CA, USA) with 0.1% bovine serum albumin (BSA) (1 h). Sections were incubated overnight (4°C) with primary antibody against MYH7 (type I fibres; mouse monoclonal A4.951, Santa Cruz, sc-53090, diluted 1:25) or MYH2 (type IIA fibres; mouse monoclonal SC71, DSHB, Iowa City, IA, USA 1:25), each combined with primary antibody against TNNT1 (rabbit polyclonal HPA058448,

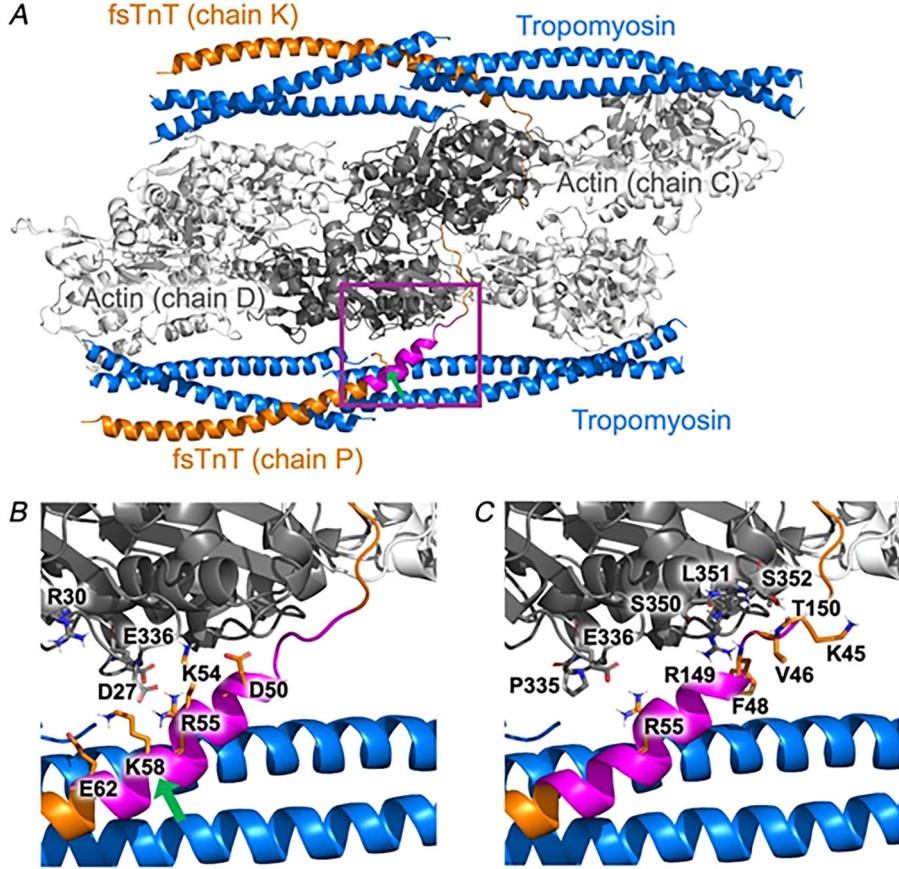

**Figure 3. Model of fsTnT in the thin filament junction**
*A*, cartoon representation of the human skeletal thin filament junction after energy minimization, showing fsTnT (orange), tropomyosin (blue) and actin (white, with the subunits interacting with fsTnT coloured in grey). The region with altered interaction patterns in WT and Lys58Gln MD trajectories is highlighted in magenta. *B* and *C*, close-up view of the fsTnT (orange liquorice) and actin (grey) residues involved in altered HB interactions (*B*) and inter-residue contacts (*C*). [Colour figure can be viewed at wileyonlinelibrary.com]

Sigma, St Louis, MO, USA, diluted 1:500), TNNT3 (rabbit polyclonal HPA037810, Sigma, diluted 1:500) or laminin (rabbit polyclonal, Sigma, L9393, diluted 1:20) in 5% goat serum with 0.1% BSA and 0.1% Triton X-100. Alexa Fluor goat anti-mouse 647 (A21237) was used as the secondary antibody for the MYHs and Alexa Fluor donkey anti-rabbit 488 (A11034) for the TNNTs and laminin (Life Technologies, 1:500 each in 10% normal goat serum). Fluorescence images were obtained with a 10× and 20× objective on a Zeiss Axio Observer 3 fluorescence microscope with a Colibri 5 LED detector, combined with Zeiss Axiocam 705 mono camera, using Zen software (Zeiss) and exported to single-channel.tiff files. The single 10× images with fibre membrane staining were then loaded into ilastik (version 1.4.0.post1, www. ilastik.org) and the software was trained to distinguish between muscle fibres and their membrane. The results were exported from ilastik as binary H5 files with the Simple Segmentation setting and analysed in Fiji (ImageJ version 1.53c with ilastik Fiji plugin, https://imagej.net/) with a macro using the analysis particle function and ROI manager tool (region of interest). Briefly, the macro automatically generated a ROI-set with individual ROIs outlining the perimeter of the muscle fibres. The ROI-set

was then visually inspected, and errors were manually corrected, before the CSAs were exported to an Excel file. Fibre type was manually assigned to the unique ID of each ROI in the Excel file (Berg et al., 2019; Schindelin et al., 2012). Fibres positive for MYH7 were considered slow twitch whilst pure or hybrid fibres positive for MYH2 were defined as fast twitch.

## Statistical analysis

Data were analysed and figures prepared in Graphpad Prism version 10.4. Various statistical tests were used and significance was set to $P < 0.05$. For histology and mavacamten (a myosin ATPase inhibitor) analyses, two-way ANOVAs with Tukey's *post hoc* test were applied (see Figs 1 and 9). For the post-translational modification search and X-ray diffraction analyses, unpaired $t$ tests with Welch correction or one-way ANOVAs with Tukey's *post hoc* test were used (see Figs 2–7). For contractile measurements, mixed effects models were run as previously published (Krivickas et al., 2011) (see Fig. 8). These mixed effects models ensured appropriate weighting of data including subjects and multiple muscle fibres per subject.

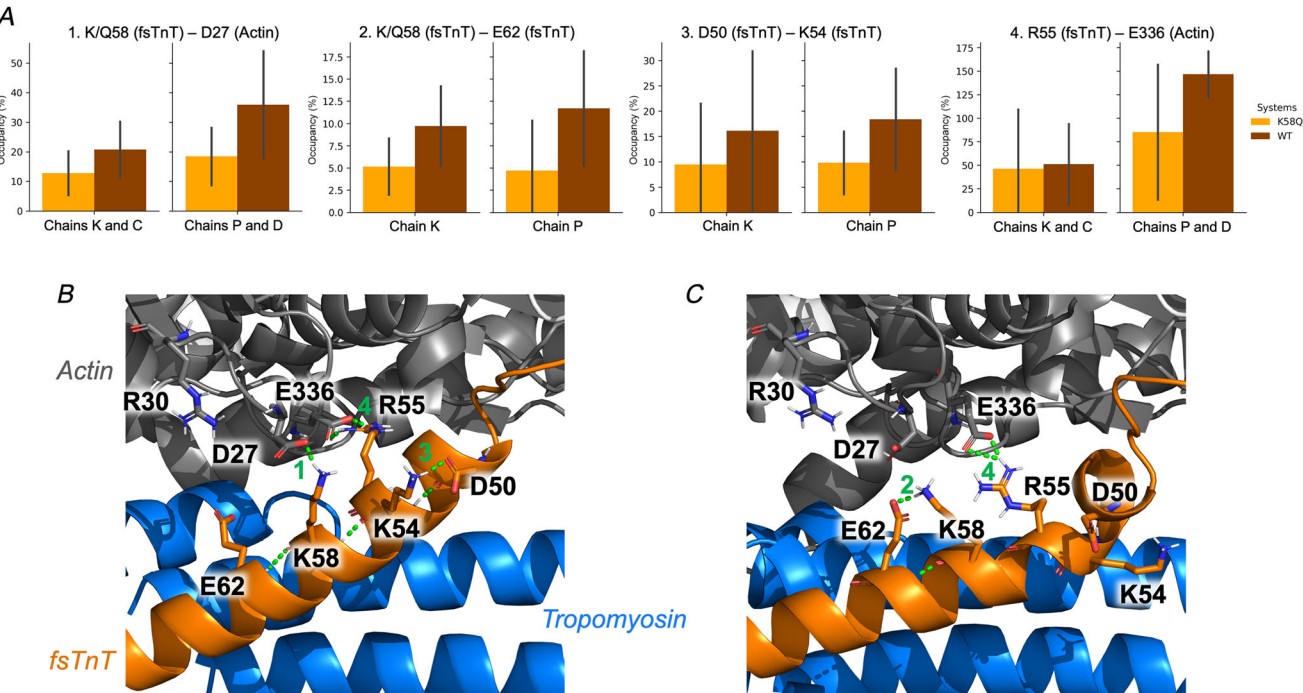

**Figure 4. Rewiring of the HB network illustrated with WT structures**
*A*, occupancy of HB interactions (calculated as the percentage of frames where a given HB interaction is observed) for selected residue pairs from fsTnT and actin. Average values calculated over five replicas are shown for Lys58Gln (orange) and WT (dark orange) MD simulations, with the standard deviation of the mean shown as a bar. Values calculated for both sets of chains (fsTnT chains K and P, and actin chains C and D) are shown for each interaction. *B* and *C*, representative structures extracted through a cluster analysis of WT trajectories to illustrate the HB interactions 1–4, which are on average stronger in WT than Lys58Gln simulations. A close-up view of residues from the P (fsTnT) and D (actin) chains is shown. [Colour figure can be viewed at wileyonlinelibrary.com]

## Results

### Histological abnormalities within muscle fibres where *TNNT1* pathogenic variants are present

We focused our attention on muscle biopsy specimens from patients with NM and *TNNT1* mutations, as well as healthy controls (Table 1). Initial quantitative histological measurements did not show any differences in fibre type proportions between patients and controls (Fig. 1*A*, *B*). However, when stratifying the patient population into 'mild' and 'severe' cases (Table 1), we found a significantly lower proportion of slow twitch muscle fibres (expressing MYH7) in severe cases than in mild cases and controls (Fig. 1*C*). Additionally, as previously observed (Abdulhaq et al., 2016; Fattahi et al., 2017; Konersman et al., 2017; Marra et al., 2015; van der Pol et al., 2014), CSAs of slow twitch fibres were smaller in patients than controls whilst CSAs of fast twitch muscle fibres were very heterogeneous but not significantly different between groups (Fig. 1*D*). When separating patients according to their phenotypes, CSAs of slow twitch myofibres were smaller in both mild and severe cases when compared to controls (Fig. 1*E*). CSAs of fast twitch fibres were greater in mild cases than in severe cases and controls

(Fig. 1*E*). The genetic variants studied here were diverse and induced single amino acid substitutions or deletions (Table 1), probably disrupting ssTnT secondary structure and/or ssTnT presence/incorporation within sarcomeres. Further qualitative histological analyses also revealed changes in the expressions of ssTnT and fsTnT among patients (Fig. 1*A*). Indeed, mild cases exhibited normal distributions while severe cases displayed an absence of ssTnT and over-expression of fsTnT, notably in slow twitch fibres (Fig. 1*A*).

All these abnormalities within muscle fibres indicate primary pathological and/or secondary adaptive responses with *TNNT1* variants.

### Abnormal post-translational modifications on TnT in the presence of *TNNT1* pathogenic variants

Based on the above findings, we sought to investigate whether further alterations would affect ssTnT and fsTnT. Hence, muscle samples were run on 6% SDS-PAGE gels, and bands for ssTnT (P13805; TNNT1_HUMAN) and fsTnT (P45378-7; TNNT3_HUMAN) subsequently excised and specifically screened for acetylation and phosphorylation using LC/MS (Sonne et al., 2023). Briefly,

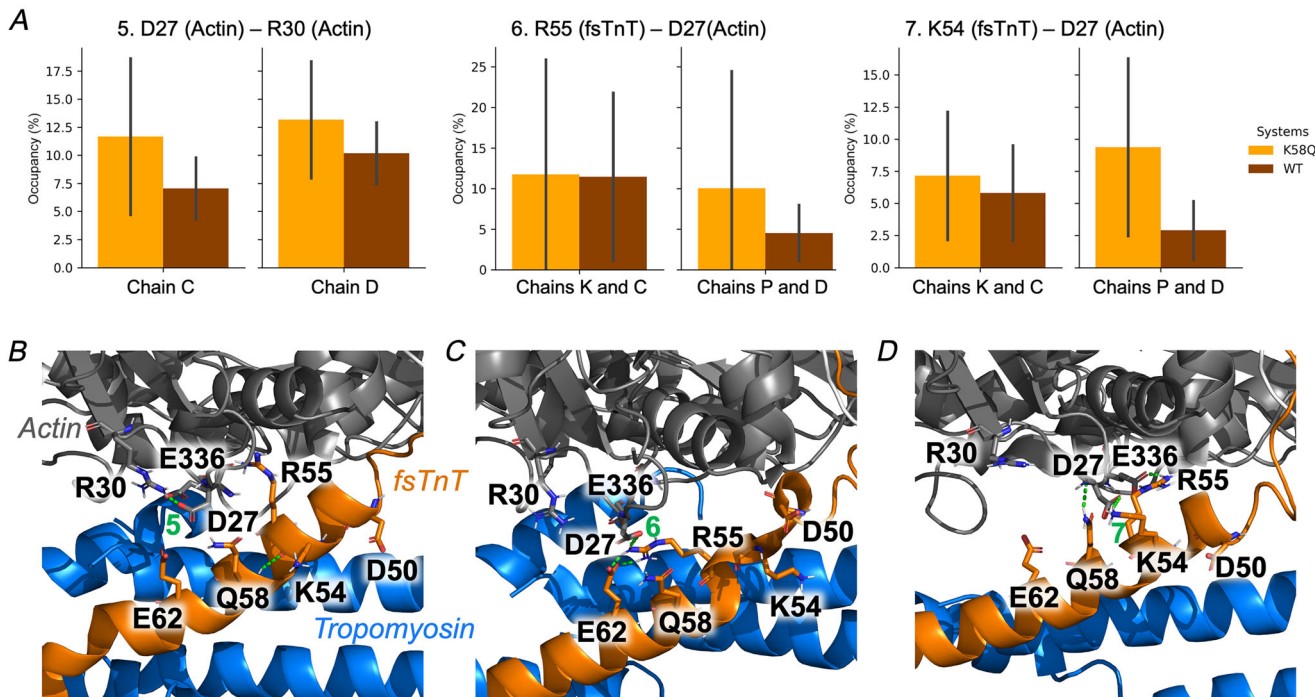

**Figure 5. Rewiring of the HB network illustrated with Lys58Gln structures**
*A*, occupancy of HB interactions (calculated as the percentage of frames where a given HB interaction is observed) for selected residue pairs from fsTnT and actin. Average values calculated over five replicas are shown for Lys58Gln (orange) and WT (dark orange) MD simulations, with the standard deviation of the mean shown as a bar. Values calculated for both sets of chains (fsTnT chains K and P, and actin chains C and D) are shown for each interaction. *B* and *C*, representative structures extracted through a cluster analysis of Lys58Gln trajectories to illustrate the HB interactions 5–7, which are on average stronger in Lys58Gln than WT simulations. A close-up view of residues from the K (fsTnT) and C (actin) chains is shown. [Colour figure can be viewed at wileyonlinelibrary.com]

peptide site intensities were normalized to the maximum peak intensities and converted to Z-scores (Sonne et al., 2023). Surprisingly, two aberrant post-translational modifications, namely reduction of both Lys58-Ac and Ser148-P, were identified in fsTnT patients compared to controls (Fig. 2*A*, *B*). When dividing the patient population into 'mild' and 'severe' cases (Table 1), the Lys58-Ac Z-score was significantly smaller in severe cases than in mild cases and controls (Fig. 2*C*). Unexpectedly, the Ser148-P Z-score was more variable in the controls than in the mild and severe cases but was significantly smaller in both sub-categories when compared to controls (Fig. 2*D*).

As a proof-of-concept, to define the functional relevance of these unusual post-translational modifications, we studied the effects of Lys58Gln (mimicking Lys58-Ac) on troponin/thin filament structure and dynamics (Fig. 3). Ser148 was not investigated as it is located in a disordered region with no solved structure. MD simulations were run in multiple 500 ns replicas on a model of the human skeletal thin filament junction containing two fsTnT segments (orange cartoon in Fig. 3*A*), eight tropomyosin segments (blue) and six actin subunits (grey). Comparison of the WT and Lys58Gln trajectories highlighted consistent changes in the HB interactions and inter-residue contacts involving TnT and actin residues in the region close to the mutation (purple in Fig. 3*A–C*). Specifically, exchanging the positively charged Lys for the neutral Gln induces a rewiring of the HB network in that region, which is rich in acidic and basic residues (Figs 4 and 5). For example, without a competing basic residue in position 58, TnT residues Lys54 and Arg55 and actin residue Arg30 can form stronger interactions with Asp27 from actin (Fig. 5), while HB interactions involving the mutated residue become weaker (Fig. 4). Albeit to a different extent, these

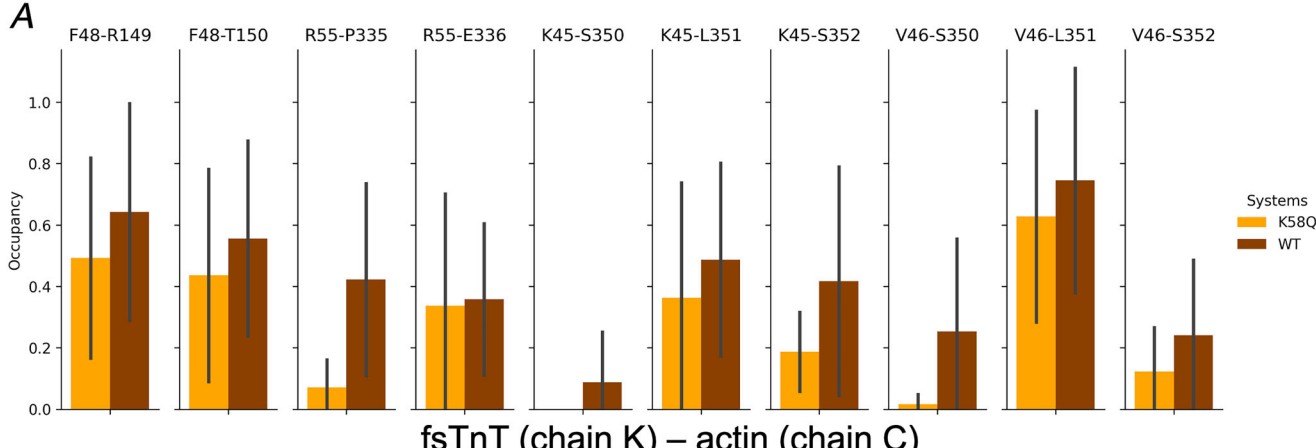

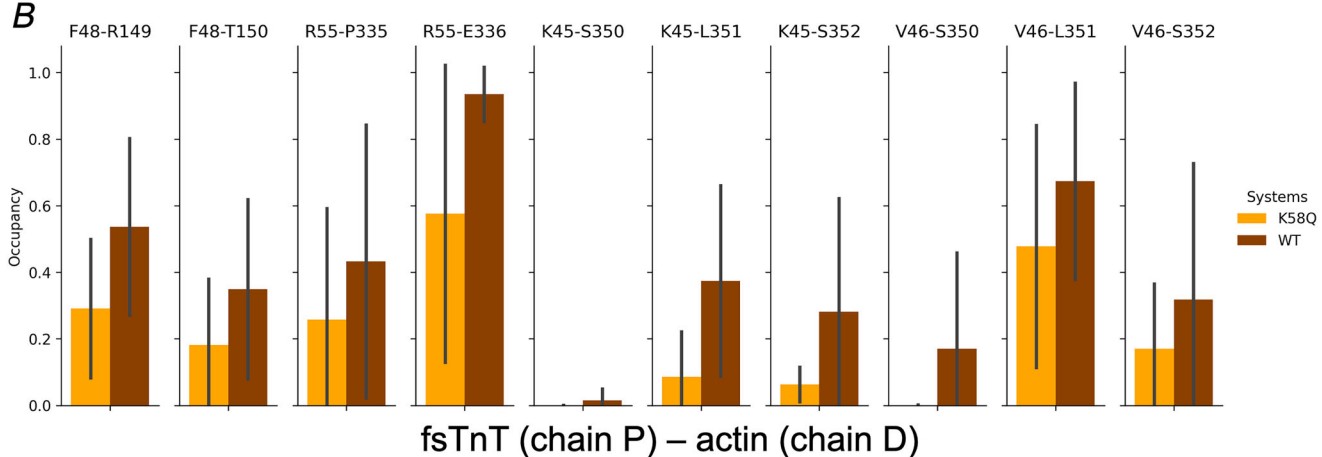

**Figure 6. Contact analysis**
*A* and *B*, occupancy of contacts (calculated as the fraction of frames where a given contact is observed) for selected residue pairs from fsTnT and actin. Average values calculated over five replicas are shown for Lys58Gln (orange) and WT (dark orange) MD simulations, with the standard deviation of the mean shown as a bar. Values calculated for fsTnT (chain K) – actin (chain C) and fsTnT (chain P) – actin (chain D) pairs are shown in *A* and *B*, respectively. [Colour figure can be viewed at wileyonlinelibrary.com]

differences were observed consistently across replicas and in both TnT segments. Analysis of changes in contacts, independently from the specific type of interaction, also showed consistent differences between WT and mutant simulations in the TnT–actin interface close to the mutation site (Figs 3*C* and 6).

Taken together, abnormal post-translational modifications affect key functional regions of TnT isoforms, making them potentially prone to altered binding to actin monomers (and/or to tropomyosin and other troponin subunits), ultimately affecting the stability or compliance of the thin filament.

### *TNNT1* pathogenic variants disrupt thin filament compliance

To experimentally characterize thin filament extensibility, we recorded X-ray diffraction patterns of control and patient muscle bundles in relaxing conditions (Chan et al., 2016) (Fig. 7*A*). We analysed two thin filament-specific meridional reflections (Fig. 7*B*), the sixth actin layer line reflection (ALL6) and the third-order meridional reflection of troponin (TN3). As patients' bundles had many small and fragile myofibres with weak intensities, equatorial and other meridional reflections were not analysed and the patients' population was not divided according to the phenotypes. Nevertheless, both the spacing of ALL6, which captures filament twist as well as axial length change (Chan et al., 2016; Hessel et al., 2022, 2024) (Fig. 7*C*), and the spacing of TN3 (Fig. 7*D*), which originates from the distance between troponin complexes (Chan et al., 2016), were greater in patients compared to controls. Finally, the intensity of the TN3 reflection, which indicates troponin order (Chan et al., 2016), was lower in patients than in controls (Fig. 7*E*).

Overall, our X-ray diffraction data are in line with increased thin filament compliance (or extensibility). In the presence of *TNNT1* pathogenic variants and abnormal fsTnT post-translational modifications, the thin filament

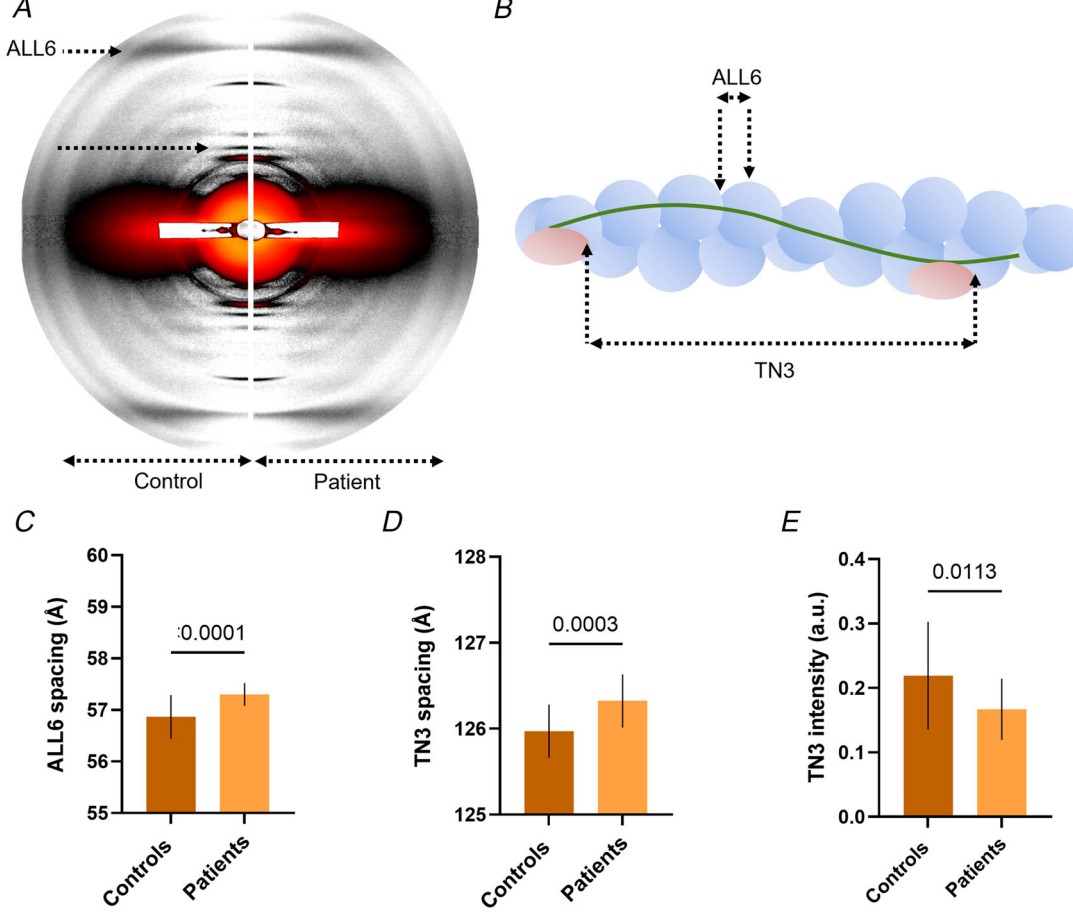

**Figure 7. Thin filament compliance**
*A*, typical X-ray diffraction patterns obtained from one control and one mild patient. *B*, the meaning of the two main reflections studied here, i.e. the 6th actin layer line, ALL6, and 3rd troponin reflection, TN3. *C*, *D* and *E*, mean and standard deviations. Unpaired *t*-tests with Welch correction were used to compare groups (level of significance *P* < 0.05). [Colour figure can be viewed at wileyonlinelibrary.com]

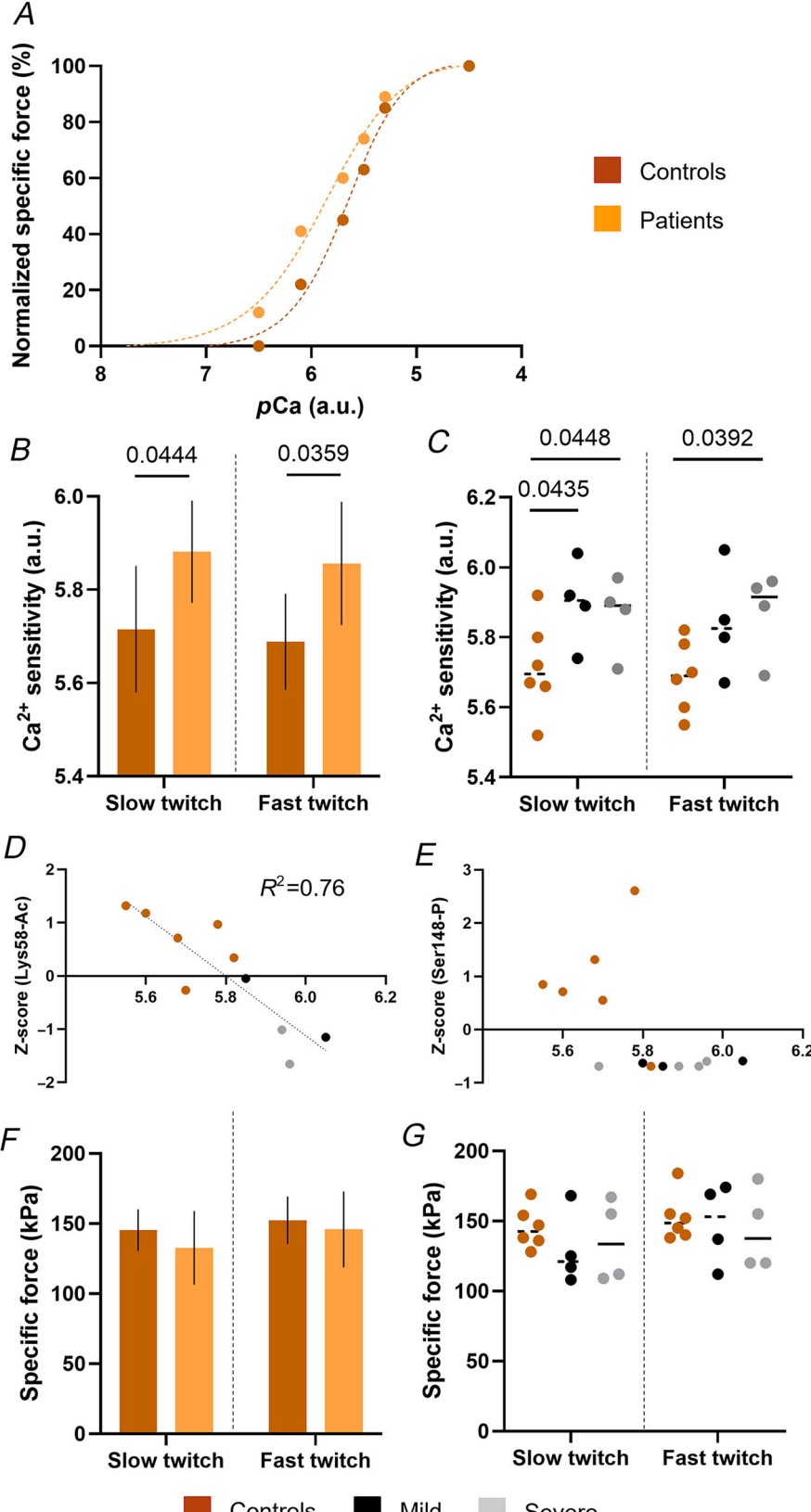

**Figure 8. Myofibre Ca²⁺ sensitivity**

*A*, representative force–*p*Ca curves from one control and one mild patient. *B* and *C*, Ca²⁺ sensitivity is displayed for controls and for all pooled patients (*B*) or for patients separated according to their phenotypes: mild or severe

(*C*). *D* and *E*, correlations between $Ca^{2+}$ sensitivity and Lys58-Ac Z-score (*D*) or Ser148-P Z-score (*E*). *F* and *G*, specific force is shown for controls and for all patients (*F*) or for mild *vs.* severe patients (*G*). Mean, standard deviations and/or individual data points (circles) are shown. A mixed effects models were used to analyse the data as previously published (Krivickas et al., 2011) (level of significance $P < 0.05$). *$P < 0.05$. Otherwise, for *D* and *E*, regression analyses were performed and the $R^2$ is presented for the only significant correlation. [Colour figure can be viewed at wileyonlinelibrary.com]

may never reach a fully relaxed state, preferring its structure following activation by $Ca^{2+}$. This is then likely to have detrimental consequences for contractility.

### *TNNT1* pathogenic variants cause a myofibre hyper-contractile state

Thin filament activation favours myosin cross-bridge formation (Gordon et al., 2000). To experimentally verify whether *TNNT1* variants promote a cellular hyper-contractile state (and relate our findings to clinical muscle symptoms), we evaluated the force-generating capacity of isolated membrane-permeabilized muscle fibres. A total of 281 myofibres were included in the analysis (minimum of 20 fibres per individual) and data were separated between slow and fast twitch fibres. $Ca^{2+}$ sensitivity was higher in patients than controls for both fibre types (Fig. 8*A*, *B*). When separating patients according to their phenotypes, in slow twitch myofibres, we found a significantly higher $Ca^{2+}$ sensitivity in mild and severe cases when compared to controls (Fig. 8*C*). In fast twitch myofibres, the higher $Ca^{2+}$

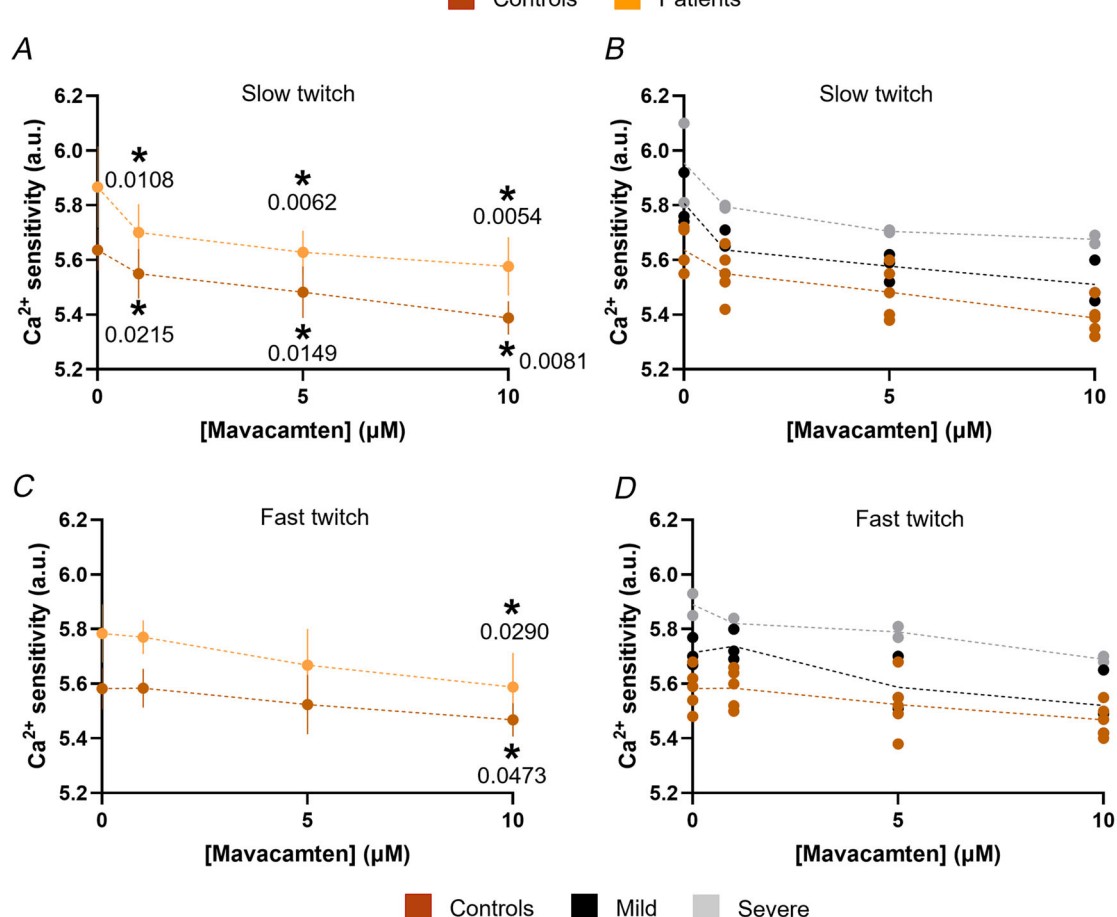

**Figure 9. Muscle fibre response to a contractile inhibitor**

*A–D*, $Ca^{2+}$ sensitivity in both slow twitch and fast twitch muscle fibres with varying concentrations of mavacamten (0, 1, 5 and 10 µM). Individual data points (circles) are shown for pooled patients or patients divided according to their phenotypes: mild or severe. Two-way ANOVAs with repeated measures and *post hoc* analyses were performed with $P < 0.05$ as level of significance (Factor 1: group of individuals – Factor 2: mavacamten concentration). *Differences when compared with 0 µM mavacamten, irrespective of the group. [Colour figure can be viewed at wileyonlinelibrary.com]

sensitivity was only present in severe cases (Fig. 8*C*). In an attempt to link these findings to the abnormal post-translational modifications observed above in fsTnT, we drew correlations between fast twitch $Ca^{2+}$ sensitivity and Lys58-Ac/Ser148-P Z-scores. Interestingly, Lys58-Ac (but not Ser148-P) was significantly and linearly related to $Ca^{2+}$ sensitivity, supporting the idea that such post-translational modification regulates contractility (Fig. 8*D*, *E*). Besides these striking changes in $Ca^{2+}$ sensitivity, no significant difference was observed in the maximum isometric specific force production (Fig. 8*F*, *G*).

Based on these findings, we tested the effects of one contractile (myosin) inhibitor known to target MYH7 (within slow twitch fibres): mavacamten (Laitila et al., 2024). Because of the amount of tissue left we focused our analyses on the $Ca^{2+}$ sensitivity of five controls and five patients only (three mild cases and two severe cases). A total of 132 individual muscle fibres were exposed to three different concentrations of mavacamten: 1, 5 and 10 μM (12–15 myofibres for each of the 10 sub-jects). As expected, in slow twitch fibres, even a small concentration of mavacamten (1 μM) was sufficient to significantly decrease $Ca^{2+}$ sensitivity in both controls and patients (Fig. 9*A*). Such decrease was further pronounced at greater concentrations of mavacamten in both controls and patients (5 and 10 μM; Fig. 9*A*). When dividing patients according to their phenotypes, we observed similar patterns in mild and severe cases (Fig. 9*B*). On the other hand, in fast twitch muscle fibres, we only detected an effect of mavacamten at a concentration of 10 μM in controls and pooled patients (Fig. 9*C*) or mild/severe cases (Fig. 9*D*). Unfortunately, at the time of our investigations, we were unable to access specific pharmacological inhibition of MYH2 and/or MYH1 (within fast twitch muscle fibres) and left this for future works.

## Discussion

In the present study, we confirm our initial hypothesis and propose a potential mechanism for the muscle phenotype observed in *TNNT1*-linked NM. Whilst contractures are often caused by muscle weakness, they may also be due to hyper-contractility, as has been observed in other forms of NM (Jain et al., 2012). Indeed, our experiments reveal that, in the presence of nonsense or missense *TNNT1* pathogenic variants, ssTnT and fsTnT expression and their phosphorylation/acetylation patterns are variable and aberrant (histology and LC/MS); the thin filament compliance is accrued (X-ray diffraction); and $Ca^{2+}$ sensitivity is increased (skinned myofibre mechanics). Taken together, these mechanisms are probably responsible for a molecular and cellular hyper-contractility, ultimately contributing to the muscle phenotype, in particular contractures, in *TNNT1*-related NM.

### Diverse patho-mechanisms in NM

In addition to *TNNT1* variants, NM has been associated with defects in genes encoding other proteins directly or indirectly involved in thin filament regulation, namely *NEB*, *ACTA1*, *TPM3*, *TPM2*, *KLHL40*, *KLHL41*, *KBTKD13*, *CFL2*, *LMOD3* and *MYPN*. More specifically, single amino acid substitutions or deficiencies in nebulin, skeletal actin, tropomyosin, Kelch repeat and BTB domain-containing protein 13 or leiomodin 3 have now been proven to inhibit thin filament activation, preventing proper myosin cross-bridge formation and muscle fibre force-generating capacity (de Winter et al., 2024; Joureau et al., 2018; Ottenheijm et al., 2009, 2011; Yuen et al., 2014). These hypo-contractile molecular and cellular events, also present in *TNNC2*-related congenital myopathy (van de Locht et al., 2021), have been related to hypotonia and muscle weakness. Residue substitutions or deficiencies in TnT, on the other hand, have the opposite biophysical effects and result in molecular and cellular hyper-contractility. This is likely to contribute to the marked muscle symptoms observed in *TNNT1*-related congenital myopathies. Similar findings have been reported with mutations in the *TNNI1* gene encoding one other troponin sub-unit, TnI, in the *ACTA1* gene, and in other congenital muscle disorders where contra-ctures also often occur (Donkervoort et al., 2024; Jain et al., 2012). These divergent patho-mechanisms in *TNNT1*-related NM are important considerations when designing possible therapeutic interventions.

*TNNT1* pathogenic variants are thought to represent a critical component of the hyper-contractile cascade of events unravelled here. Nevertheless, as ssTnT is mainly expressed in slow twitch muscle fibres, the mutations cannot explain why $Ca^{2+}$ sensitivity is also increased when ssTnT is lacking or in fast twitch myofibres where fsTnT is predominant (and ssTnT absent), suggesting additional and potentially complex pathophysiological mechanisms. Our analyses indicate that fsTnT regions rich in lysine and serine, known to be essential for thin filament activation, have unusual levels of acetyl-ation and phosphorylation (Fig. 2*C*). Thus, in addition to the genetic mutations, Lys58-Ac and Ser148-P may locally destabilize fsTnT and affect the binding of other essential thin filament proteins such as actin and tropomyosin, overall impairing protein compliance. Such a (secondary) adaptative and detrimental role of aberrant post-translational modifications on contractile proteins has already been demonstrated in the context of other congenital myopathies related to *RYR1* mutations (Sonne et al., 2023).

## Targeting hyper-contractility to reverse contractures

In the present study, muscle fibre hyper-contractility is proposed as a key determinant of muscle symptoms in NM due to *TNNT1* pathogenic variants. Considering this background, lowering thin filament activation and/or subsequent myosin cross-bridge formation are two potentially promising pharmacological therapeutic approaches (Claassen et al., 2023; Jungbluth et al., 2017). $Ca^{2+}$ desensitizers are still at an early developmental stage, whilst myosin inhibitors have attracted more attention recently (Claassen et al., 2023; Jungbluth et al., 2017). Historically, blebbistatin is the most widely used myosin class II inhibitor but is not appropriate and toxic. Among the other myosin inhibitors, FDA-approved mavacamten (also known as MYK-461 or Camzyos) decreases the rate of phosphate release and stabilizes the myosin inactive OFF state and is more effective in slow twitch muscle fibres expressing the $\beta$/slow myosin heavy chain (Claassen et al., 2023). Preclinical MPH-220, another small molecule known to alter the actin binding cleft, is, on the other hand, more efficient in fast twitch muscle fibres expressing the myosin heavy chain IIA and IIX (Claassen et al., 2023). Similarly, preclinical EDG-5506 selectively targets fast twitch myofibres by disrupting the phosphate release rate (Claassen et al., 2023). As *TNNT1* pathogenic variants have impacts on both slow and fast twitch muscle fibres, our findings emphasize that a combination of mavacamten and fast myosin inhibitor may be considered the preferential choices for *TNNT1*-related congenital myopathies. Further *in vivo* studies specifically assessing these are needed.

## Conclusions

Here, we aimed to obtain a better understanding of the molecular and cellular mechanisms underlying the muscle dysfunction in patients with *TNNT1* variants. Interestingly, our findings indicate that the mutations result in unusual post-translational modifications, alter thin filament compliance through hyper-activation and increased $Ca^{2+}$ sensitivity, ultimately inducing a hyper-contractile phenotype in both slow and fast twitch muscle fibres. These mechanisms are probable contributors to the contractures seen in *TNNT1*-mutated patients and may be potentially amenable to treatment with contractile inhibitors.

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

## Additional information

### Data availability statement

The MS proteomics data have been deposited with the ProteomeXchange Consortium via the PRIDE partner repository (Perez-Riverol et al., 2022) with the dataset identifier PXD054504. The rest of the data are shown in the figures but, if needed, these can be made available upon request.

### Competing interests

C.T.A.L. is an employee of Novo Nordisk A/S. This position began after their work on this paper and the position had no influence on the results or conclusions drawn. M.W.L. is the CEO and founder of Diverge Translational Science Laboratory, which works with numerous companies in the areas of therapeutic development and testing. His work at Diverge is unrelated to the therapeutic options discussed here and his position had no influence on the results or conclusions drawn. A.L.H. is an owner of Accelerated Muscle Biotechnologies Consultants LLC, which performed the X-ray data reduction and analysis, but services rendered were not linked to outcome or interpretation.

### Author contributions

J.L., C.T.A.L., C.A.C., A.F. and J.O. contributed to the study conception and design. Material preparation, data collection and analysis were performed by J.L., C.T.A.L., A.L.H., G.P., A.H.L., C.F., M.W.L., C.A.C., H.J., K.F.M., A.F. and J.O. The first draft of the manuscript was written by J.L. and J.O. and all authors commented on all versions of the manuscript and approved of the submitted version.

### Author Translational Perspective

In the present study, we aimed to determine how *TNNT1* variants linked to nemaline myopathy induce muscle cell dysfunction. Our findings indicate that these variants are associated with a molecular remodelling at the sarcomere level consisting of stiff thin filament and hyper-contractility. These

results give, for the first time, a strong rationale for using existing contractile inhibitors (such as mavacamten) to cure *TNNT1*-related nemaline myopathy. The results also warrant the design of novel inhibitors specifically targeting the stiff thin filament.

## Funding

This work was generously funded by the Novo Nordisk Foundation (grant agreement number NNF21OC0070539) to J.O.; and by the Engineering and Physical Sciences Research Council (EP/T518086/1) to K.F.M. and A.F. Additionally, the Engineering and Physical Sciences Research Council allowed us to use time on HPC granted via the UK High-End Computing Consortium for Biomolecular Simulation, HECBioSim (http://hecbiosim.ac.uk) (grant no. EP/X035603/1). G.P. is a member of the European Reference Network for Neuromuscular Diseases – Project ID No. 870177.

## Acknowledgements

The authors thank Dr Hiroyuki Iwamoto and Thomas Nyegaard Beck for their assistance in many of the experiments outlined in the present paper. The X-ray experiments were performed under approval of the SPring-8 Proposal Review Committee (2022B1107). Additionally, the authors thank Dr Anders Karlsen for AI ilastik training and for developing the macro for CSA measurements with ImageJ (FiJi). MS analyses were performed by the Proteomics Research Infrastructure (PRI) at the University of Copenhagen (UCPH), supported by the Novo Nordisk Foundation (grant agreement no. NNF19SA0059305).

## Keywords

contracture, congenital myopathy, force, skeletal muscle, troponin

## Supporting information

Additional supporting information can be found online in the Supporting Information section at the end of the HTML view of the article. Supporting information files available:

**Peer Review History**

