## [Peer Review History · The Journal of Physiology]

Pathogenic TNNT1 variants are associated with aberrant thin filament compliance and myofibre hyper-contractility

Jenni Iaitila, Christopher T A Lewis, Anthony L Hessel, Guido Primiano, Aurelio Hernandez-Lain, Chiara Fiorillo, Michael Lawlor, Coen Ottenheijm, Heinz Jungbluth, Ka Fu Man, Arianna Fornili, and Julien Ochala

DOI: 10.1113/JP288109

Corresponding author(s): Julien Ochala (julien.ochala@sund.ku.dk)

The following individual(s) involved in review of this submission have agreed to reveal their identity: Jian-Ping Jin (Referee #2)

Review Timeline:

Submission Date:	13-Nov-2024
Editorial Decision:	02-Jan-2025
Revision Received:	19-Mar-2025
Editorial Decision:	11-Apr-2025
Revision Received:	11-Apr-2025
Accepted:	15-Apr-2025

Senior Editor: *Peying Fong*

Reviewing Editor: *Kevin Murach*

Transaction Report:

Dear Dr Ochala,

Re: JP-RP-2024-288109 "Pathogenic TNNT1 variants are associated with aberrant thin filament compliance and myofibre hyper-contractility" by Jenni Iaitila, Christopher Lewis, Anthony L Hessel, Guido Primiano, Aurelio Hernandez-Lain, Chiara Fiorillo, Michael Lawlor, Coen Ottenheijm, Heinz Jungbluth, Ka Fu Man, Arianna Fornili, and Julien Ochala

Thank you for submitting your manuscript to The Journal of Physiology. It has been assessed by a Reviewing Editor and by 2 expert referees and we are pleased to tell you that it is potentially acceptable for publication following satisfactory major revision.

REVISION CHECKLIST:

We look forward to receiving your revised submission.

Yours sincerely,

Peying Fong
Senior Editor
The Journal of Physiology

REQUIRED ITEMS

- Author photo and profile. First or joint first authors are asked to provide a short biography (no more than 100 words for one author or 150 words in total for joint first authors) and a portrait photograph. These should be uploaded and clearly labelled together in a Word document with the revised version of the manuscript. See Information for Authors for further details.
- You must start the Methods section with a paragraph headed Ethical Approval. If experiments were conducted on humans, confirmation that informed consent was obtained, preferably in writing, that the studies conformed to the standards set by the latest revision of the Declaration of Helsinki and that the procedures were approved by a properly constituted ethics committee, which should be named, must be included in the article file. If the research study was registered (clause 35 of the Declaration of Helsinki), the registration database should be indicated, otherwise the lack of registration should be noted as an exception (e.g. The study conformed to the standards set by the Declaration of Helsinki, except for registration in a database). For further information see: <https://physoc.onlinelibrary.wiley.com/hub/human-experiments>.
- The Journal of Physiology funds authors of provisionally accepted papers to use the premium BioRender site to create high resolution schematic figures. Follow this link and enter your details and the manuscript number to create and download figures. Upload these as the figure files for your revised submission. If you choose not to take up this offer, we require figures to be of similar quality and resolution. If you are opting out of this service to authors, state this in the Comments section on the Detailed Information page of the submission form. The link provided should only be used for the purposes of this submission. Authors will be charged for figures created on this premium BioRender account if they are not related to this manuscript submission.
- Please upload separate high-quality figure files via the submission form.
- Your paper contains Supporting Information of a type that we no longer publish, including supplementary tables and figures. Any information essential to an understanding of the paper must be included as part of the main manuscript and figures. The only Supporting Information that we publish are video and audio, 3D structures, program codes and large data

files. Your revised paper will be returned to you if it does not adhere to our Supporting Information Guidelines.

- Papers must comply with the Statistics Policy: https://jp.msubmit.net/cgi-bin/main.plex?form_type=display_requirements#statistics.

In summary:

- If $n \leq 30$, all data points must be plotted in the figure in a way that reveals their range and distribution. A bar graph with data points overlaid, a box and whisker plot or a violin plot (preferably with data points included) are acceptable formats.

- If $n > 30$, then the entire raw dataset must be made available either as supporting information, or hosted on a not-for-profit repository, e.g. FigShare, with access details provided in the manuscript.

- 'n' clearly defined (e.g. x cells from y slices in z animals) in the Methods. Authors should be mindful of pseudoreplication.

- All relevant 'n' values must be clearly stated in the main text, figures and tables.

- The most appropriate summary statistic (e.g. mean or median and standard deviation) must be used. Standard Error of the Mean (SEM) alone is not permitted.

- Exact p values must be stated. Authors must not use 'greater than' or 'less than'. Exact p values must be stated to three significant figures even when 'no statistical significance' is claimed.

- Please include an Abstract Figure file, as well as the Figure Legend text within the main article file. The Abstract Figure is a piece of artwork designed to give readers an immediate understanding of the research and should summarise the main conclusions. If possible, the image should be easily 'readable' from left to right or top to bottom. It should show the physiological relevance of the manuscript so readers can assess the importance and content of its findings. Abstract Figures should not merely recapitulate other figures in the manuscript. Please try to keep the diagram as simple as possible and without superfluous information that may distract from the main conclusion(s). Abstract Figures must be provided by authors no later than the revised manuscript stage and should be uploaded as a separate file during online submission labelled as File Type 'Abstract Figure'. Please also ensure that you include the figure legend in the main article file. All Abstract Figures should be created using BioRender. Authors should use The Journal's premium BioRender account to export high-resolution images. Details on how to use and access the premium account are included as part of this email.

Reviewing Editor's comments:

Your work has been evaluated by two expert reviewers. Both viewed the work as quite influential and will be a valuable addition to the literature, but major revision will be required. Please pay particular attention to comments regarding the statistics and age-matching of participant samples, as well as concerns regarding fiber type proportions.

Senior Editor's comments:

The initial review of your manuscript is now complete. Two expert referees and the Reviewing Editor consider its potential for impact in advancing mechanistic understanding of TNNT1 mutation-associated myopathies to be high. However, you will read that there are nonetheless aspects that require additional attention. These include experimentation (and possibly better design incorporating tighter age-matching) as well as details of the chosen statistical approach, as raised by Referee 1. Referee 2 identifies many limiting gaps in understanding. If addressed thoroughly, the findings would vastly strengthen the study's conclusions.

I encourage you to consider and address fully the suggestions offered by both Referees when preparing your revised manuscript.

Referee #1:

Please see attachment [JP-RP-2024-288109_Referee 1 Review Attachment 1.pdf]

Referee #2:

This work investigated several myopathic TNNT1 mutations with structural and functional characterizations, some of which have not been studied for the molecular and cellular mechanisms underlying the muscle dysfunction. The authors presented an interesting finding that the TNNT1 mutations result in post-translational modifications in fast TnT encoded by the TNNT3 gene. The authors concluded that the loss of TNNT1 function altered thin filament compliance and produced a hyper-contractile phenotype in both slow and fast twitch muscle fibers. While the new findings are interesting and could contribute to the understanding of the pathogenesis of TNNT1 myopathies from various point mutations, there are some concerns that

require the authors' attention.

- 1) The study emphasizes on using structural studies including molecular dynamic simulation but missed the opportunity to more thoroughly characterize the histopathology and contractility of patient muscle samples of representative TNNT1 mutations.
- 2) Fig. 1 shows there appears an adaptive hypertrophy of fsTnT fibers in TNNT1 myopathies, which was confirmed in Fig. 5 in isolated fibers. This phenotype should be quantified and compared among representative TNNT1 mutations, which may contribute to the functional changes in isolated muscle fibers due to shifting to more fast fiber contents.
- 3) In the single muscle fiber force studies, only type I MHC was examined thus the ratio of MHC isoforms in mixed MHC type fibers cannot be evaluated but needs to be considered for making an informative conclusion.
- 4) It is also unclear whether and how the single fibers verified for myosin isoform contents were matched to the contractility data.
- 5) The hypothesis that the altered thin filament compliance in TNNT1 mutant fibers was through hyper-activation and increased Ca(2+) sensitivity needs to be supported by a feasible mechanism and testable model. It is unclear how increased maximum activation would make the resting thin filament more compliant. Could adaptive fast fiber type shift contribute to this phenotype?
- 6) The finding that Ca(2+) sensitivity was higher in TNNT1 myopathy patient muscles than controls for both fiber types (Fig. 6A) needs to be analyzed against the actual mutations, especially for the severe loss of slow TnT function mutants (eg, E180X, E203X and E221X) vs the other mutants. Force-pCa curves should be provided.
- 7) The fsTnT posttranslational changes should be analyzed and discussed in the context of slow TnT mutations.
- 8) The observation that a myosin motor inhibitor, mavacamten, decreased Ca(2+) sensitivity specifically in slow twitch muscle fibers (Fig. 6C) is interesting but needs more in-depth investigation for underlying mechanisms. An adaptive change in myosin isoform contents may contribute to why it only affected slow twitch fibers. Mavacamten dose response curves of fast and slow fibers from patients with specific TNNT1 mutation and controls may provide additional information. To support the conclusion, a testable mechanism linking myosin activity and thin filament Ca(2+) sensitivity needs to be proposed.
- 9) The finding that the TNNT1 mutations result in post-translational modifications in fast TnT encoded by the TNNT3 gene provides evidence of adaptive changes and should be discussed for pathogenic and functional significances.
- 10) The authors concluded that the contractures seen in TNNT1 myopathy patients may be treated with contractile inhibitors needs more careful consideration as muscle weakness is the main abnormality of TNNT1 patients.

END OF COMMENTS

In their manuscript, “Pathogenic TNNT1 variants and associated with aberrant thin filament compliance and myofiber hyper-contractility” the authors investigate potential mechanisms explaining altered thin filament excitability often associated with nemaline myopathies (NM). The multi-institutional investigative team bring to bear an impressive collection of research approaches to characterize proteinaceous, structural and functional modifications associated with genetic mutations that lead to nemaline myopathies. They conclude that these mutations lead to protein substitutions, deletions and posttranslational modification to the troponin T protein, which in turn, contribute to enhanced compliance of the thin filament and a pathogenic inability to become fully inactive. Their findings build on previous observations while delivering valuable mechanistic insight. While the myosin inhibitor Mavacamten did not resolve hypercontractility preferentially demonstrated in this patient group (MyHC II dominant), but it has greater efficacy in MyHC I. Nevertheless it would have been quite valuable to see if MyHC II inhibition would have helped resolve NM based differences in functional outcomes at the fiber level.

Major:

The patient group includes a dramatic range of ages (<1 to 51 yrs) not reflected in the controls (21 to 55 yrs). This would almost certainly impact variation and means of fiber type distribution, morphology and force. Were secondary analyses performed on a subset of data where samples were more reasonably matched for age of participant?

Lines 238 – 241 Statistical approach is notably light on details. For analyses including multiple observations per subject (single fiber mechanics and Mavacamten treatment), were mixed models used to account for inter-subject variability? It seems as though statistics were measured as a mean of all fibers per subject with post hoc testing to identify fiber specific differences. Would it not have made more sense to assess outcome measures within fiber types across groups with a mixed model to account for variation within individuals across fibers?

Minor:

Line 115: the PRIDE identifier PXD054504 does not appear to be associated with an active dataset.

Line 204: Specific force or Tension are typically used to distinguish absolute force from force expressed relative to cross section. Please adjust accordingly.

Line 289: Can the authors provide a reasonable explanation for why other meridional reflections were weak only in patients? This seems like a potentially important finding.

Line 618: The compression of Z-scores in patients for phosphorylation of Ser148 is quite dramatic. Variation in patients, by contrast, is more typical. Were correlations assessed between mean phosphoenrichment per patient and structural or functional outcome measures?

March 19th, 2025

Dear Editor, Journal of Physiology

Please find enclosed our revised manuscript. We are grateful to the Editor and reviewers for constructive criticisms.

Yours Sincerely,

Julien Ochala

REVIEWING EDITOR

Please pay particular attention to comments regarding the statistics and age-matching of participant samples, as well as concerns regarding fiber type proportions.

R. Please see specific responses to Reviewers 1 and 2 for these.

SENIOR EDITOR

You will read that there are nonetheless aspects that require additional attention. These include experimentation (and possibly better design incorporating tighter age-matching) as well as details of the chosen statistical approach, as raised by Referee 1. Referee 2 identifies many limiting gaps in understanding. If addressed thoroughly, the findings would vastly strengthen the study's conclusions.

R. Please see specific responses to Reviewers 1 and 2 for these.

REVIEWER 1

It would have been quite valuable to see if MyHC II inhibition would have helped resolve NM based differences in functional outcomes at the fiber level.

R. We agree with the reviewer that such experiments would strengthen our manuscript. However, we have not been able to access specific pharmacological inhibitors of MYH2 and/or MYH1 (within fast twitch muscle fibres). To acknowledge this limitation, we have now added a sentence at the end of the Results section (paragraph titled 'TNNT1 pathogenic variants cause a myofibre hyper-contractile state').

The patient group includes a dramatic range of ages (<1 to 51 yrs) not reflected in the controls (21 to 55 yrs). This would almost certainly impact variation and means of fiber type distribution, morphology and force. Were secondary analyses performed on a subset of data where samples were more reasonably matched for age of participant?

R. We agree with the reviewer. This age difference may cause some unknown consequences. Unfortunately, it is ethically very difficult to obtain tissue from healthy children. To make sure the reader is aware of this limitation, we have added a sentence in the legend of Table 1. Additionally, and in line with other comments from reviewer #1 and #2 and the editor, we have separated patients according to their phenotypes: mild versus severe. Severe patients are children's cases.

Lines 238 – 241 Statistical approach is notably light on details. For analyses including multiple observations per subject (single fiber mechanics and Mavacamten treatment), were mixed models used to account for inter-subject variability? It seems as though statistics were measured as a mean

of all fibers per subject with post hoc testing to identify fiber specific differences. Would it not have made more sense to assess outcome measures within fiber types across groups with a mixed model to account for variation within individuals across fibers?

R. Please accept our apologies for this potential confusion or misunderstanding. We have now tried to be clear and explain the various analyses we have performed. Accordingly, we have now changed the statistics paragraph at the end of the Methods section. We have also changed the legends of all the figures.

Line 115: the PRIDE identifier PXD054504 does not appear to be associated with an active dataset.

R. Please accept our apologies for that mistake. It is now public.

Line 204: Specific force or Tension are typically used to distinguish absolute force from force expressed relative to cross section. Please adjust accordingly.

R. Done.

Line 289: Can the authors provide a reasonable explanation for why other meridional reflections were weak only in patients? This seems like a potentially important finding.

R. In line with the reviewer's comment, we have now added a sentence in the Results section (paragraph 'TNNT1 pathogenic variants disrupt thin filament compliance'). Indeed, one potential qualitative explanation for that may relate to the presence of small and fragile muscle fibres in the patient muscle bundles we tested.

Line 618: The compression of Z-scores in patients for phosphorylation of Ser148 is quite dramatic.

R. We agree with the reviewer that finding is unexpected and surprising. We have now added this observation to the Results section (paragraph 'Abnormal post-translational modifications on TnT in the presence of TNNT1 pathogenic variants'). However, we have decided not to expand as the underlying mechanisms remain unclear and would be too speculative in the present manuscript.

Were correlations assessed between mean phospho enrichment per patient and structural or functional outcome measures?

R. In line with the reviewer's comment, we have now attempted to correlate the unusual post-translational modifications with our contractile measurements. Interestingly and unexpectedly, we have now observed a clear correlation between Ca^{2+} sensitivity and Lys58-Ac. We have now added a

few sentences in the Results section (paragraph 'TNNT1 pathogenic variants cause a myofibre hypercontractile state') and we have also changed Figure 8.

REVIEWER 2

The study missed the opportunity to more thoroughly characterize the histopathology and contractility of patient muscle samples of representative TNNT1 mutations.

R. Accordingly, we have now better explained this. Please see the Results section and the new paragraph 'Histological abnormalities within muscle fibres where TNNT1 pathogenic variants are present' and the new Figure 1.

Fig. 1 shows there appears an adaptive hypertrophy of fsTnT fibers in TNNT1 myopathies, which was confirmed in Fig. 5 in isolated fibers. This phenotype should be quantified and compared among representative TNNT1 mutations, which may contribute to the functional changes in isolated muscle fibers due to shifting to more fast fiber contents.

R. As mentioned in the answer of the previous comment, we have now a new paragraph in the results section ('Histological abnormalities within muscle fibres where TNNT1 pathogenic variants are present') and the new Figure 1. This should help the reader better understand the complex remodelling that occurs histologically.

In the single muscle fiber force studies, only type I MHC was examined thus the ratio of MHC isoforms in mixed MHC type fibers cannot be evaluated but needs to be considered for making an informative conclusion. It is also unclear whether and how the single fibers verified for myosin isoform contents were matched to the contractility data.

R. Please accept our apologies for this potential confusion. To identify the type of fibres (slow-twitch versus fast-twitch), we used IHC after the contractile measurements. Pure or hybrid fibres positive for MYH7 were considered slow twitch whilst negative myofibres were defined as fast twitch (as previously published by our group in the Journal of Physiology, Laitila et al., 2024). To make sure the reader is aware of that, we have added a sentence in the Methods section, paragraph 'Single muscle fibre force production'.

The hypothesis that the altered thin filament compliance in TNNT1 mutant fibers was through hyperactivation and increased Ca^{2+} sensitivity needs to be supported by a feasible mechanism and testable model. It is unclear how increased maximum activation would make the resting thin filament more compliant. Could adaptive fast fiber type shift contribute to this phenotype?

R. Our experiments have taken into account fibre types (see Figure 8). Hence, it is unlikely that any fibre type shift would explain our results. However, we believe our data (taking into account fibre types) show that the trigger in slow twitch fibres is the mutation leading to aberrant compliance/extensibility of the thin filament which in turn modifies myosin binding to actin and contractility. In fast twitch fibres, the trigger is potentially the aberrant post-translational modifications as indicated by our MD analyses and correlations in Figure 8. The text has been changed accordingly.

The finding that Ca^{2+} sensitivity was higher in TNNT1 myopathy patient muscles than controls for both fiber types (Fig. 6A) needs to be analyzed against the actual mutations, especially for the severe loss of slow TnT function mutants (eg, E180X, E203X and E221X) vs the other mutants. Force-pCa curves should be provided.

R. We agree with the reviewer. To account for potential differences between mutations, we have now separated between mild and severe mutations. We have now amended the results section and figures accordingly.

The fsTnT posttranslational changes should be analyzed and discussed in the context of slow TnT mutations.

R. These unusual post-translational modifications appeared to be affecting fsTnT only (not ssTnT). In line with the reviewer's comment, we have now separated patients according to the mutations/severity of the phenotypes. Results and related figures have then been modified accordingly.

The observation that a myosin motor inhibitor, mavacamten, decreased Ca^{2+} sensitivity specifically in slow twitch muscle fibers (Fig. 6C) is interesting but needs more in-depth investigation for underlying mechanisms. An adaptive change in myosin isoform contents may contribute to why it only affected slow twitch fibers. Mavacamten dose response curves of fast and slow fibers from patients with specific TNNT1 mutation and controls may provide additional information.

R. In line with the reviewer's comment we have now run additional experiments including various concentrations of mavacamten. Please see Results section, paragraph 'TNNT1 pathogenic variants cause a myofibre hyper-contractile state' and the new Figure 9. We agree with the reviewer, this is still a proof of concept and should open up new research/manuscripts focusing on the exact underlying mechanisms.

The finding that the TNNT1 mutations result in post-translational modifications in fast TnT encoded by the TNNT3 gene provides evidence of adaptive changes and should be discussed for pathogenic and functional significances.

R. Discussion has been changed accordingly.

The authors concluded that the contractures seen in TNNT1 myopathy patients may be treated with contractile inhibitors needs more careful consideration as muscle weakness is the main abnormality of TNNT1 patients.

R. We have tried to be cautious and changed the text accordingly. We now talk about muscle symptoms rather than contractures/weakness.

Dear Dr Ochala,

Re: JP-RP-2025-288109R1 "Pathogenic TNNT1 variants are associated with aberrant thin filament compliance and myofibre hyper-contractility" by Jenni Iaitila, Christopher Lewis, Anthony L Hessel, Guido Primiano, Aurelio Hernandez-Lain, Chiara Fiorillo, Michael Lawlor, Coen Ottenheijm, Heinz Jungbluth, Ka Fu Man, Arianna Fornili, and Julien Ochala

Thank you for submitting your manuscript to The Journal of Physiology. It has been assessed by a Reviewing Editor and by 2 expert referees and we are pleased to tell you that it is acceptable for publication following satisfactory revision.

REVISION CHECKLIST:

We look forward to receiving your revised submission.

Yours sincerely,

Peying Fong
Senior Editor
The Journal of Physiology

REQUIRED ITEMS

- We invite you to include a Translational Perspective paragraph in your manuscript. This should be included in the main body of the manuscript after the Acknowledgements. It should describe the wider translational implications of the work, in plain English, for a broad scientific audience. Please use the following guidelines to prepare a Translational Perspective of your paper: https://jp.msubmit.net/cgi-bin/main.plex?form_type=display_requirements#authortranspersp. The Translational Perspective should not exceed 250 words in total and should be presented as a single paragraph. Abbreviations and technical terms must be defined as briefly and simply as possible the first time they are used, unless they are generally/easily understood, e.g. ECG, HIV/AIDS, K⁺ channel. Use language that can be understood by scientists or clinicians with a general knowledge of the topic addressed. Ensure the paragraph includes the hypothesis tested in the paper and accurately reflects the findings of the paper and the implications for future research. Please state the word count of the Translational Perspective paragraph.

- Papers must comply with the Statistics Policy: https://jp.msubmit.net/cgi-bin/main.plex?form_type=display_requirements#statistics.

In summary:

- If $n \leq 30$, all data points must be plotted in the figure in a way that reveals their range and distribution. A bar graph with data points overlaid, a box and whisker plot or a violin plot (preferably with data points included) are acceptable formats.
- If $n > 30$, then the entire raw dataset must be made available either as supporting information, or hosted on a not-for-profit repository, e.g. FigShare, with access details provided in the manuscript.
- 'n' clearly defined (e.g. x cells from y slices in z animals) in the Methods. Authors should be mindful of pseudoreplication.
- All relevant 'n' values must be clearly stated in the main text, figures and tables.
- The most appropriate summary statistic (e.g. mean or median and standard deviation) must be used. Standard Error of the Mean (SEM) alone is not permitted.
- Exact p values must be stated. Authors must not use 'greater than' or 'less than'. Exact p values must be stated to three significant figures even when 'no statistical significance' is claimed.

EDITOR COMMENTS

Reviewing Editor:

Thank you for your responsiveness to the reviewers. Your work has been re-evaluated and is potentially acceptable for publication.

Please see 'Required Items' above.

Senior Editor:

Comments for Authors to ensure the paper complies with the Statistics Policy (Required):

Statistics accompanying figural data are presented without specifying p values to 3 significant figures. Figures 4, 5, and 6 use the standard error of the mean rather than the standard deviation.

The Authors are commended to The Journal of Physiology's Statistics Policy and advised to revise to comply fully with this policy.

Comments to the Author:

Review of your revised manuscript, "Pathogenic TNNT1 variants are associated with aberrant thin filament compliance and myofibre hyper-contractility", is now complete. The attached evaluations of the two original referees, as well as the summary statement from the Reviewing Editor, indicate this version to be much improved. They concur that its potential for influencing clinical practice, development of therapies, and the field overall to be strong.

However, a few points requiring your attention before a final decision can be rendered. While it can indeed be appreciated that, in response to Referee 1, the details of statistical analyses are now more clearly presented, they do not fully comply with the journal's published Statistics Policy. I commend you specifically to points pertaining to presentation of error, which should be as standard deviation, rather than standard error of the mean as shown in figures 4, 5, and 6. Please note also that p should be presented as exact values (to three significant figures, with the exception of cases in which $p < 0.001$) and not using ">" or "<". I expect that you will be able to make these corrections readily.

Thank you for submitting to The Journal of Physiology. I look forward to receiving the final version of your manuscript.

REFeree COMMENTS

Referee #1:

The authors have satisfied this reviewers concerns. Congratulations on a nice collection of work and a well-written manuscript.

Referee #2:

The revised manuscript has addressed my previous concerns.

END OF COMMENTS

April 11th, 2025

Dear Editor, Journal of Physiology

Please find enclosed our revised manuscript. We believe we have amended the text according to the journal' guidelines. As you will see, we have now:

1. Included a Translational Perspective paragraph;
2. Specified p values (3 digits) for all figures;
3. Included the standard deviations for Figures 4, 5, and 6.

If for any reason, we have missed some of the guidelines, we would of course change the text accordingly.

Yours Sincerely,

Julien Ochala

Dear Dr Ochala,

Re: JP-RP-2025-288109R2 "Pathogenic TNNT1 variants are associated with aberrant thin filament compliance and myofibre hyper-contractility" by Jenni Iaitila, Christopher T A Lewis, Anthony L Hessel, Guido Primiano, Aurelio Hernandez-Lain, Chiara Fiorillo, Michael Lawlor, Coen Ottenheijm, Heinz Jungbluth, Ka Fu Man, Arianna Fornili, and Julien Ochala

We are pleased to tell you that your paper has been accepted for publication in The Journal of Physiology.

Yours sincerely,

Peying Fong
Senior Editor
The Journal of Physiology

If you would like to receive our 'Research Roundup', a monthly newsletter highlighting the cutting-edge research published in The Physiological Society's family of journals (The Journal of Physiology, Experimental Physiology, Physiological Reports, The Journal of Nutritional Physiology and The Journal of Precision Medicine: Health and Disease), please click this link, fill in your name and email address and select 'Research Roundup':
<https://www.physoc.org/journals-and-media/membernews>

- You can help your research get the attention it deserves! Check out Wiley's free Promotion Guide for best-practice recommendations for promoting your work at: www.wileyauthors.com/eeo/guide. You can learn more about Wiley Editing Services which offers professional video, design, and writing services to create shareable video abstracts, infographics, conference posters, lay summaries, and research news stories for your research at: www.wileyauthors.com/eeo/promotion.

EDITOR COMMENTS

Senior Editor:

Comments to the Author:

Thank you for addressing the details raised in the last cycle of review. At this time, your manuscript is ready for final acceptance. Congratulations on an excellent study!